# Difficulties of Eating and Masticating Solid Food in Children with Spinal Muscular Atrophy—Preliminary Study [note 1]

**DOI:** 10.3390/nu17152561

**Published:** 2025-08-06

**Authors:** Ewa Winnicka, Adrianna Łabuz, Zbigniew Kułaga, Tomasz Grochowski, Piotr Socha

**Affiliations:** 1Department of Otorhinolaryngology, Audiology and Phoniatrics, The Children’s Memorial Health Institute, 04-730 Warsaw, Poland; a.labuz@ipczd.pl (A.Ł.); t.grochowski@ipczd.pl (T.G.); 2Department of Public Health, The Children’s Memorial Health Institute, 04-730 Warsaw, Poland; z.kulaga@ipczd.pl; 3Department of Gastroenterology, Hepatology, Nutrition Disorders and Pediatrics, The Children’s Memorial Health Institute, 04-730 Warsaw, Poland; p.socha@ipczd.pl

**Keywords:** spinal muscular atrophy, SMA, tongue strength, dysphagia, TOMASS-C, IOPI, bulbar function, pediatric neuromuscular disorders, mastication, solid food intake

## Abstract

**Background:** Spinal muscular atrophy (SMA) is a neuromuscular disorder that frequently affects bulbar function, including feeding and swallowing. Although disease-modifying therapies have improved motor outcomes, little is known about the persistence of oromotor difficulties, particularly with regard to solid food intake. **Objective**: This study aimed to evaluate mastication and swallowing performance in children with SMA undergoing treatment, and to investigate the association between tongue strength and feeding efficiency. **Methods:** Twenty-two children with SMA types 1–3 were assessed using the Test of Masticating and Swallowing Solids in Children (TOMASS-C) and the Iowa Oral Performance Instrument (IOPI). Key TOMASS-C outcomes included the number of bites, chewing cycles, swallows, and total eating time. Tongue strength was measured in kilopascals. **Results:** Most participants showed deviations from age-specific normative values in at least one TOMASS-C parameter. Tongue strength was significantly lower than reference values in 86% of participants and correlated negatively with all TOMASS-C outcomes (*p* < 0.001). Children with weaker tongue pressure required more swallows, more chewing cycles, and longer eating times. **Conclusions**: Despite pharmacological treatment, children with SMA experience persistent difficulties in eating solid foods. Tongue strength may serve as a non-invasive biomarker for bulbar dysfunction and support dietary decision-making and therapeutic planning.

## 1. Introduction

Spinal muscular atrophy (SMA) is a genetically determined, progressive neuromuscular disorder caused by mutations in the SMN1 gene, which encodes the survival motor neuron (SMN) protein. This mutation leads to the premature degeneration of motor neurons and, consequently, muscle weakness and atrophy [1]. The natural course of the disease is classified into five types, based on the age of onset and the severity of motor symptoms [2]. Due to its progressive nature, SMA results not only in neuromuscular impairment but also in structural abnormalities of the spine, joints, and craniofacial region, often necessitating respiratory and nutritional support [3]. In types 0, 1, and 2, SMA can significantly reduce life expectancy, while in types 3 and 4, although life span may be preserved, patients typically experience decreased mobility, loss of independence, and a markedly reduced quality of life [4].

Degeneration of bulbar motor neurons contributes to difficulties in breathing, swallowing, eating, and speaking [5]. Weakness of muscles innervated by cranial nerves (V, VII, IX, X, XI, XII), along with structural changes such as temporomandibular joint contractures and malocclusion, can lead to bulbar dysphagia and dysarthria [6]. The severity of these symptoms depends on the SMA type and disease duration. In the most severe cases, such as SMA type 0, affected neonates often require mechanical ventilation and enteral feeding from birth [7,8]. In untreated children with SMA type 1, enteral nutrition is typically introduced within the first year of life due to malnutrition or aspiration, often accompanied by the need for oral secretion suctioning [9]. According to McGrattan et al., 34% of children (mean age 4.4 ± 3.1 years) required suctioning of secretions, and 39% received enteral nutrition [5].

Children with SMA type 1 also experience considerable difficulty in consuming solid foods, including an inability to chew, fatigue during meals, frequent choking, short and ineffective feeding sessions, wet breathing, and coughing during meals [10,11]. Among patients with SMA types 2 and 3, the most commonly reported feeding difficulties include trouble placing food into the mouth, limited mouth opening, oral hypersensitivity, choking and coughing during meals, prolonged eating time, aspiration, chewing difficulties, and poor weight gain [12,13]. The average age of onset for such issues is reported to be 6.5 years (range: 0–16.5 years) [14].

The introduction of disease-modifying therapies—nusinersen, onasemnogene abeparvovec, and risdiplam—has transformed the clinical course of SMA [15,16,17]. These therapies differ in their mechanism of action and route of administration, and many pediatric patients are now treated with one or more of them. Improvements have been observed in motor abilities, respiratory function, and oral feeding, regardless of treatment modality [18,19,20]. However, the evolving phenotype of SMA in the era of treatment has rendered previous clinical profiles partially outdated. While improvements in bulbar function are reported, data on the specific nature of persistent chewing and feeding challenges in treated patients remain limited [21,22].

The aim of this study is to assess the presence and characteristics of chewing and feeding difficulties in children with SMA receiving treatment, with a specific focus on the impact of tongue strength on these challenges [23].

## 2. Materials and Methods

### 2.1. Study Design and Participants

This was an observational study involving 22 children diagnosed with spinal muscular atrophy (SMA)**,** all aged over 4 years. Participants were at different stages of disease progression and had received various types of disease-modifying treatments, initiated at different points in their lives. At the time of assessment, all children were at least partially orally fed and had been referred for evaluation due to feeding difficulties reported by caregivers or healthcare professionals.

Children were recruited from multiple regions across Poland and were under the ongoing care of various clinical centers, ensuring diversity in clinical profiles and geographic background. The data presented in this article reflect a baseline functional assessment of feeding and oromotor performance, conducted prior to the initiation of any targeted therapeutic intervention. The outcomes of those interventions are being analyzed separately and will be presented in a subsequent publication.

The study received ethical approval from the Bioethics Committee of the Children’s Memorial Health Institute in Warsaw, Poland (approval number: 63/KBE/2019). Written informed consent was obtained from the legal guardians of all participants prior to inclusion in the study. All participants were assessed individually and in the presence of their parents.

### 2.2. Assessment Tools and Outcome Measures

The food used for testing was classified as Level 7a according to the International Dysphagia Diet Standardisation Initiative (IDDSI) framework. IDDSI is a globally recognized framework that provides standardized terminology and definitions for texture-modified foods and thickened liquids used in dysphagia management. It ensures consistency in assessment and dietary recommendations across clinical settings [24].

IDDSI Level 7a—Regular (Easy to Chew) is characterized as follows:It includes regular, everyday foods that are naturally soft in texture and require minimal chewing.Foods at this level do not require modifications in particle size but should be easy to chew and swallow safely.It is intended for individuals who do not need texture-modified diets but may have mild chewing difficulties or fatigue.Foods should be moist, tender, and not hard, tough, crunchy, or dry.

This level is often used in early dysphagia or mild oral-motor impairments, where full normal textures might be too challenging, but puree or minced diets are unnecessarily restrictive.

#### 2.2.1. Test of Masticating and Swallowing Solids in Children (TOMASS-C)

Swallowing and mastication of solids were evaluated based on video recordings taken during the consumption of a Gran Pavesi Salati™ cracker (IDDSI level 7a).

Four factors were assessed:The number of discrete bites required to finish the cracker.The number of masticatory cycles (defined as up-and-down jaw movements, excluding rotary jaw and tongue movements used to clear oral residue).The number of swallows (observed via the movement of the thyroid cartilage).The total duration of eating (measured with a digital stopwatch from the first bite to the final swallow) [25].

These variables were selected as key indicators of oromotor function and efficiency during solid food intake. Children consumed the cracker and were instructed to eat it as quickly as possible, meaning they were expected to eat efficiently without unnecessary pauses for talking or looking around the room. The goal was to encourage them to focus on the act of eating itself. However, this did not mean that we created an atmosphere of haste or pressure. The tests were conducted in the presence of the children’s parents, who were also given the opportunity to eat the same type of cracker. This approach was intended to reduce any tension related to the recording situation and to help the children feel more at ease during the assessment.

Every movement aimed at biting off a piece of the cracker was evaluated. In some cases, the effectiveness of biting off pieces was limited to just a few crumbs. High and atypical scores in this part of the test reflect a lack of skill in performing this task.

Each cycle typically includes one opening and one closing movement of the jaw. Chewing cycles occur in a rhythmic and repetitive pattern during mastication. The jaw not only moves up and down but may also shift side to side (in a lateral or rotary motion), especially in more mature chewing patterns. The main purpose is to mechanically break down food and mix it with saliva to form a cohesive bolus for swallowing. While the cycle is defined by jaw movement, effective chewing also involves coordinated movements of the tongue and cheeks to position food between the teeth. Chewing cycles can vary in speed and number depending on food texture, age, neuromuscular function, and the individual’s oral-motor control. According to TOMASS-C guidelines movements used for clearing residue from the oral cavity (e.g., with the tongue) or repositioning food without crushing it are not considered part of the chewing cycle.

In neuromuscular diseases, the primary issue is reduced ability to clear the oral cavity and pharynx effectively. As a result, patients often rely on compensatory strategies, which include repeated attempts to initiate swallowing of a single bolus and fragmentary swallowing. These strategies are reflected in an increased number of swallows per single bolus. In accordance with the TOMASS-C guidelines, we counted each swallow initiation, which in our understanding reflects the difficulties in clearing the pharynx and oral cavity.

The video analysis began at the moment the cracker touched the child’s lips and ended when the child said their name or the word “done”, indicating completion of the task.

Reference ranges for each variable were based on 95% confidence intervals (CIs) from age-matched normative TOMASS-C data [25]. Notably, a value of zero in any domain was interpreted not as a valid performance outcome, but rather as the result of non-completion or failure to initiate the task. Such cases were retained in the dataset to reflect test feasibility and real-world behavioral presentation.

#### 2.2.2. Iowa Oral Performance Instrument (IOPI)

To evaluate maximal tongue muscle strength, we used the IOPI [26]. Measurements were performed using an air-filled bulb placed between the tongue and the hard palate, connected to a pressure transducer. The pressure generated by pressing the bulb against the palate was recorded in kilopascals (kPa)**.** We used the maximum pressure (Pmax) function for assessment. The measurement was repeated three times, and the highest value was recorded. The results were compared with reference values established for typically developing children and adolescents [27].

### 2.3. Statistical Analysis

Normality in the data was assessed by examination of histogram, skewness, and kurtosis and Kolmogorov–Smirnoff test. The results of swallowing and masticating tests were compared with age- and sex-specific TOMASS-C reference ranges (95% confidence intervals) for the parameters under the study. The relationship between swallowing, masticating, and tongue strength was analyzed using Spearman’s rank correlation. A *p* value less than 0.05 was regarded as statistically significant. A statistical analysis was performed using SAS statistical software version 9.4 (SAS Institute Inc., Cary, NC, USA).

## 3. Results

### 3.1. Study Sample

Twenty-two children diagnosed with SMA were included in the study. All participants were over 4 years of age at the time of assessment, with ages ranging from 49 to 190 months (mean = 83.7 months, SD = 32.5). The cohort included individuals with SMA type 1 (*n* = 17), type 2 (*n* = 3), and type 3 (*n* = 2).

Most children were receiving nusinersen, either as monotherapy or in combination with other disease-modifying therapies, such as risdiplam and/or onasemnogene abeparvovec. Several participants had received multiple pharmacological therapies throughout the course of their disease. Two participants had been treated with branaplam during earlier phases of their therapy, although the drug is no longer in clinical use.

With regard to respiratory support, 12 children required non-invasive ventilation (NIV), which was typically administered during the night for durations ranging from 4 to 12 h. The remaining 10 participants did not use proactive ventilatory support. None of the participants were invasively ventilated, and no child had a tracheostomy at the time of the assessment.

In terms of nutritional status, all participants were at least partially orally fed. The majority (*n* = 19) were exclusively orally fed, while three children received more than 50% of their nutritional intake via enteral feeding, indicating varying degrees of oromotor and swallowing dysfunction.

Table 1 reflects the clinical heterogeneity of the study population in terms of age, SMA type, therapeutic approach, respiratory support, and feeding status.

### 3.2. Objective Measures TOMASS-C

Figure 1, Figure 2, Figure 3 and Figure 4 illustrates outcomes recorded for 22 participants compared to TOMASS-C normative references range. The green bars represent the normative 95% confidence intervals (CIs). Each dot represents an individual participant’s score. Two participants (P12 and P16) recorded a count of zero, signifying either a failure to initiate the task or an incomplete test; these cases are also marked in red and may reflect functional, behavioral, or sensory difficulties.

### 3.3. Number of Bites

Figure 1 illustrates the number of bites. A total of 17 participants demonstrated bite counts within or below the normative range. Three participants (P4, P9, and P20) exceeded the upper bound of the reference range and are marked in red.

### 3.4. Number of Chewing Cycles

Figure 2 illustrates the number of chewing cycles recorded for each participant. Of the 22 participants, 14 exhibited chewing cycle counts within or below the expected normative range (blue markers). Four participants (P2, P4, P14, and P20) recorded values that exceeded the upper bound of the reference range. Participants P9 and P18 had borderline values, yet they still exceeded the normative thresholds.

### 3.5. Number of Swallows

Figure 3 displays the number of swallowing events recorded for 22 participants. Only one participant (P6) demonstrated a swallowing frequency that fell within or below the normative range (blue marker). The remaining 21 participants recorded values exceeding the upper bound of the normative reference (red markers).

### 3.6. Total Eating Time

Each dot in Figure 4 represents the observed total time (in seconds) to complete the eating task. Only six participants (P1, P6, P7, P11, P15, and P21) completed the task in a timeframe within or below the expected reference range. The remaining 16 participants exceeded the reference interval, including two participants (P12 and P16) who recorded a time of zero.

### 3.7. Objective Measures IOPI

The scatter plot (Figure 5) presents individual maximum tongue pressure measurements (in kPa) obtained using the Iowa Oral Performance Instrument (IOPI). Each dot represents one participant.

Blue dots indicate results that fall within the age- and sex-specific reference range (green shaded area).Red dots indicate results that fall outside the reference range, suggesting decreased or abnormal tongue muscle strength.The green shaded rectangles represent normative intervals based on published reference data.

Only three participants (P1, P3, P7) demonstrated results that were appropriate for their age and sex.

### 3.8. Results of Statistical Analysis

Spearman’s rank correlations (r) and the levels of statistical significance (*p*) were as follows:Number of bites versus tongue strength: r = −0.72; *p* = 0.0003;Number of chewing cycles versus tongue strength: r = −0.72; *p* = 0.0004;Number of swallows versus tongue strength: r = −0.83; *p* < 0.0001;Eating time (second) versus tongue strength: r = −0.81; *p* < 0.0001.

All correlations were negative and statistically significant.

## 4. Discussion

Our findings confirmed that despite the use of disease-modifying therapies, the following statements applied:Tongue strength in children with SMA, as measured by IOPI, was significantly lower than in their healthy peers.Performance in the TOMASS-C deviated from normative values across several domains. Two participants (P12, P16) were unable to complete the test, indicating potential difficulties with task initiation or regulation. In the “number of bites” domain, 23% showed signs of feeding difficulty. For “chewing cycles”, 36% demonstrated abnormal values, with some exceeding the normative range, possibly due to inefficient mastication. Almost all participants (95%) had atypical “number of swallows”, and 73% showed prolonged “total eating time”, indicating delayed oral processing.Tongue strength showed a statistically significant relationship with feeding efficiency across all domains measured by TOMASS-C. A reduction in tongue strength increased the likelihood of poorer performance.

Colot et al. (2024), in their study comparing tongue strength in children with SMA and healthy controls, arrived at conclusions similar to ours [28]. Trucco et al. (2023) [29] studied children with SMA 2 and 3 and found tongue weakness in only 50% of the children studied, mainly those classified as non-ambulatory. Ambulatory patients demonstrated near-normative results. In our cohort, 86% of participants had tongue strength below normative values for their age and sex.

The results for all four TOMASS-C domains also diverged from those of Trucco et al. [29]. In our study, participants more frequently exhibited difficulties related to feeding and swallowing. Trucco et al. [29] reported issues in the “number of chewing cycles,” “number of swallows,” and “total eating time” only among non-ambulatory patients; ambulatory individuals generally scored within normative ranges. A study by van der Heul et al. (2022), involving older participants (ages 13–65, SMA types 2 and 3) also confirmed challenges primarily in these three domains [30].

These discrepancies can likely be attributed to differences in participant group characteristics. Our sample included children with SMA types 1, 2, and 3 (with 77% representing SMA type 1), whereas the studies by Trucco et al. [29] and van der Heul et al. [30] included only SMA types 2 and 3. This suggests that SMA subtype significantly influences assessment results, and that children with SMA type 1—despite receiving disease-modifying therapy—constitute a population requiring particularly careful evaluation and management of bulbar function.

However, in terms of TOMASS-C domains, similar to Trucco et al. [29] and van der Heul et al. [30], our study also demonstrated that the most frequent feeding difficulties were related to the “number of chewing cycles,” “number of swallows,” and “total eating time.”

Statistical analysis revealed that tongue strength was significantly correlated with all four TOMASS-C parameters. In the study by Trucco et al. [29], such a relationship was found only for “number of swallows” and “total eating time.” Considering that their cohort generally scored better than ours, one may hypothesize that tongue muscle weakness first manifests in an increased number of swallows and chewing cycles, and in prolonged eating time, before affecting the “number of bites.” This hypothesis requires further investigation.

The tongue plays a critical role in both the oral and pharyngeal phases of swallowing. During the oral phase, it controls bolus positioning, maintains it between the molars for mastication, and subsequently propels the bolus posteriorly toward the oropharynx to initiate the reflexive pharyngeal phase. In this phase, the tongue contributes to the glossopharyngeal seal, increasing pharyngeal pressure and narrowing the pharyngeal space in coordination with other muscles. The synergy of tongue and associated musculature facilitates pharyngeal clearance and bolus redirection into the esophagus. In SMA—as well as in other neuromuscular disorders—patients often experience difficulties with oral and pharyngeal clearance. It is possible that reduced tongue strength in the oral phase also compromises the pharyngeal clearance mechanism. Van der Heul et al. (2023) identified an association between mastication difficulties and swallowing problems in SMA types 2 and 3 [31]. This supports the need for further studies to confirm such a relationship. If validated, IOPI and TOMASS-C could serve as valuable tools for monitoring bulbar function.

Currently, VFSS (Videofluoroscopic Swallowing Study) is the recommended assessment for feeding and swallowing difficulties in children with SMA [3]. Despite receiving treatment, these children still require ongoing monitoring due to the risk of pneumonia, bronchiectasis, atelectasis, mucus plugging, and bronchial wall thickening. However, VFSS is often unavailable in many centers due to its high cost and the need for specialized equipment and trained personnel. Assessing the predictive value of clinical tests like IOPI and TOMASS-C in relation to dysphagia in neuromuscular diseases could greatly improve proactive care and allow clinicians to refer only those who truly require further instrumental evaluation. Mano et al., in a study on adults, suggested that reduced tongue strength may serve as a predictor of neuromuscular disease progression [32].

Tongue strength values may also help determine the most suitable diet for a given patient. Umemoto et al. (2020) reached this conclusion after comparing maximum tongue pressure with diet recommendations based on VFSS results in adults with neurological and neuromuscular disorders [33]. While their study focused on adults, we believe it is worthwhile to consider the application of IOPI and TOMASS-C in guiding dietary decisions or assessing the need for enteral feeding. In our study, participant P4, who had results far below normative values across all tests, is now receiving enteral support. At the time of assessment, however, they were fed exclusively orally. The objective test results supported the caregivers in making the decision to initiate gastrostomy-based nutrition.

We are currently observing varied patient profiles: children treated pre-symptomatically, those treated in early symptomatic stages, and children with structural changes resulting from established disease. It is well established that disease-modifying therapies can improve or maintain muscle strength, but they cannot reverse abnormal structural development, entrenched compensatory patterns, or maladaptive motor habits. Specialized Speech–Language Therapy is therefore essential to maximize the development and functionality of bulbar skills. Our study contributes to identifying which specific aspects of bulbar function require intervention and how these influence feeding effectiveness.

The strength of our study is its inclusion of various methods to evaluate swallowing problems in patients with SMA, which allowed us to identify a relationship.

The limitation of our study is the relatively small sample size, which restricts the generalizability of our findings. However, as SMA is a very rare disorder (SMA occurs with an estimated frequency of approximately 1 in 10,000 live births) [34], we believe that our observation of a strong correlation between tongue strength and impaired solid food intake merits attention. The sample size did not allow us to analyze data with regard to SMA type. We also did not focus on treatment type or age at treatment initiation.

However, our findings provide important information for clinicians and for future studies. The observations from our cohort must be replicated in other samples before broader conclusions can be drawn.

## 5. Conclusions

Children with spinal muscular atrophy (SMA), despite receiving disease-modifying therapy, continue to demonstrate significant impairments in tongue strength and orofacial function compared to their healthy peers. These deficits are strongly associated with reduced efficiency in all four domains assessed by the TOMASS-C test: number of bites, chewing cycles, swallows, and total eating time.

Our findings suggest that weakened tongue strength may serve as an early indicator of emerging difficulties in feeding and swallowing, particularly in children with SMA type 1. The strong correlation between tongue pressure and TOMASS-C outcomes highlights the potential clinical utility of these simple, non-invasive tools for monitoring bulbar function.

Given the limitations of access to instrumental assessments such as VFSS, clinical tests like IOPI and TOMASS-C may offer valuable support for early detection, dietary planning, and decision-making regarding the need for enteral nutrition.

We hope that our proposed approach will be used in the future to help determine which treatment strategies for SMA yield the best outcomes in terms of improving or maintaining oral feeding abilities.

Future research should focus on validating these tools in larger, more diverse cohorts and on exploring their predictive value in the context of disease progression and therapeutic outcomes. Targeted speech and swallowing interventions remain essential to address persistent bulbar dysfunction and support nutritional health in this vulnerable population.

## Figures and Tables

**Figure 1 nutrients-17-02561-f001:**
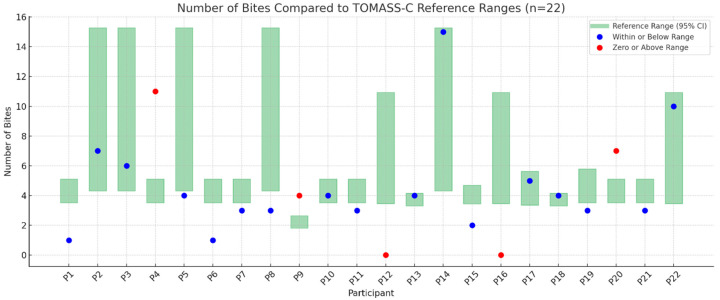
Number of bites observed in individual participants (P1–P22) compared to the TOMASS-C reference range (95% confidence interval).

**Figure 2 nutrients-17-02561-f002:**
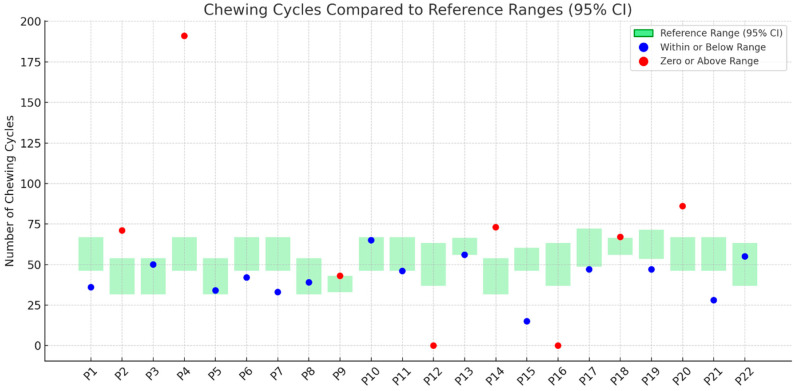
Chewing cycles performed by individual participants (P1–P22) compared to the TOMASS-C reference range (95% confidence interval).

**Figure 3 nutrients-17-02561-f003:**
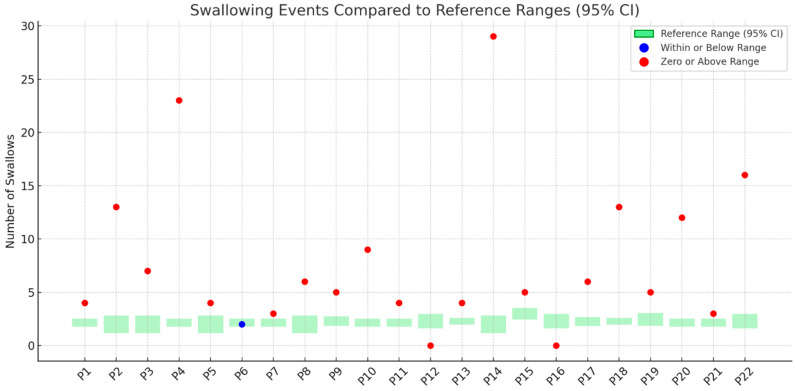
Number of swallowing events recorded for individual participants (P1–P22) compared to the TOMASS-C reference range (95% confidence interval).

**Figure 4 nutrients-17-02561-f004:**
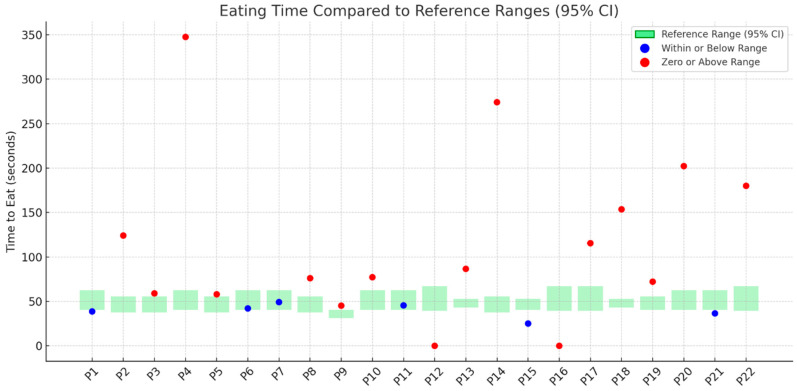
Time required to complete the eating task among individual participants (P1–P22) compared to the TOMASS-C reference range (95% confidence interval).

**Figure 5 nutrients-17-02561-f005:**
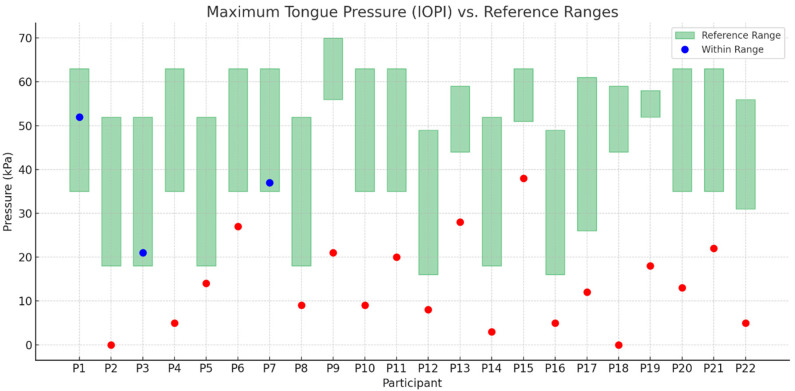
Maximum tongue pressure (IOPI) values compared to reference ranges for each participant. Red dots indicate values outside the normal range for age and sex.

**Table 1 nutrients-17-02561-t001:** Characteristics of study participants.

Participant	Age (Months)	SMA Type	Treatment	Ventilatory Support	Feeding Mode
P1	87	3	risdiplam, nusinersen	none	oral
P2	51	1	nusinersen, onasemnogen	NIV 7 h	oral
P3	60	1	branaplam, onasemnogen	none	oral
P4	80	1	nusinersen	NIV 9 h	oral
P5	48	1	nusinersen, onasemnogen	NIV 6 h	oral
P6	92	1	nusinersen	NIV 8 h	oral
P7	94	1	nusinersen	none	oral
P8	54	2	nusinersen	none	oral
P9	190	1	risdiplam	NIV 5 h	oral
P10	88	1	risdiplam	none	oral
P11	76	2	nusinersen	none	oral
P12	57	1	nusinersen, onasemnogen, risdiplam	NIV 4 h	oral
P13	116	1	nusinersen	NIV 8 h	oral
P14	57	1	risdiplam, onasemnogen	none	>50% enteral
P15	120	3	nusinersen	none	oral
P16	49	1	nusinersen, onasemnogen	NIV 11 h	oral
P17	77	1	branaplam, onasemnogen, nusinersen	NIV 12 h	oral
P18	109	1	nusinersen	none	>50% enteral
P19	112	1	nusinersen	none	oral
P20	82	1	nusinersen, onasemnogen	NIV 8–10 h	>50% enteral
P21	76	2	nusinersen	NIV 8 h	oral
P22	67	1	nusinersen	none	oral

## Data Availability

The data supporting the findings of this study are not publicly available due to privacy and ethical restrictions, but may be provided by the corresponding author upon reasonable request and approval by the appropriate ethics committee.

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
