# Peer review of "Difficulties of Eating and Masticating Solid Food in Children with Spinal Muscular Atrophy—Preliminary Study"

_nutrients, 2025, doi:10.3390/nu17152561_

Round 1

Reviewer 1 Report

Comments and Suggestions for Authors

Dear Editor and Authors,

I reviewed this study titled "Difficulties of eating and masticating solid food in children with spinal muscular atrophy – preliminary study" and I find it interesting but it needs some improvements - I have some questions:

  1. The number of subjects is small but this is possibly related to the rarity of the disease. Nevertheless it is a limitation and it needs to be reported!!
  2. The children came from different regions, different hospitals, different doctors so this opens up a big bias in terms of "diversity" of treatment and management aproaches!! Where the patients all have a uniform treatment algorithm applied or was is a bid hodge podge!!
  3. What kinds of food (7a) where used for testing? Was it only the cracker?
  4. Where the children istructed to eat normally or did they rush to finish chewing to say "done"!!
  5. There is variability on the treatment regimes/medication used. Do the authors assess this as a potential bias? Do all work via similar pathways and mechanism?
  6. Was any mouth movement considered a bite and assessed?

Thank you for giving me the oportunity to review this work.

Comments on the Quality of English Language

Needs only minor editing as it is generally a well written and understood manuscript.

Reviewer 2 Report

Comments and Suggestions for Authors

This paper evaluated masticating and swallowing function in pediatric patients with SMA using TOMASS and IOPI. Similar studies are scarce, making this a significant paper. However, there are several issues and comments.

1) The definition of the chewing cycle and its method of measurement are unclear. Please provide further explanation.

2) In children, the movement of the thyroid cartilage is difficult to observe visually. Especially in patients with neurological disorders, abnormal posture may also make it challenging to count swallows. Please describe how the authors overcame this issue.

3) Was the TOMASS evaluation conducted by a single evaluator or through discussion among multiple evaluators?

4) It is questionable whether young children aged 4-5 year-old can follow instructions such as “eat as quickly as possible” or “press the air-filled bulb firmly against the hard palate.” Therefore, it is questionable whether the short eating time and low tongue pressure are due to impaired swallowing function or an inadequate response to instructions.

5) I think that "three participants (P3, P14, and P19)" in P5,L5 should be “three participants (P4, P9, and P20)”.

6) Type I, II, III in P4,L4 should be type 1, 2, 3
